# Microscopic Interactions of Melatonin, Serotonin and Tryptophan with Zwitterionic Phospholipid Membranes

**DOI:** 10.3390/ijms22062842

**Published:** 2021-03-11

**Authors:** Jordi Martí, Huixia Lu

**Affiliations:** 1Department of Physics, Technical University of Catalonia-Barcelona Tech, 08034 Barcelona, Spain; 2School of Pharmacy, Shanghai Jiaotong University, Shanghai 200240, China; huixia.lu@sjtu.edu.cn

**Keywords:** melatonin, serotonin, tryptophan, phospholipid membrane

## Abstract

The interactions at the atomic level between small molecules and the main components of cellular plasma membranes are crucial for elucidating the mechanisms allowing for the entrance of such small species inside the cell. We have performed molecular dynamics and metadynamics simulations of tryptophan, serotonin, and melatonin at the interface of zwitterionic phospholipid bilayers. In this work, we will review recent computer simulation developments and report microscopic properties, such as the area per lipid and thickness of the membranes, atomic radial distribution functions, angular orientations, and free energy landscapes of small molecule binding to the membrane. Cholesterol affects the behaviour of the small molecules, which are mainly buried in the interfacial regions. We have observed a competition between the binding of small molecules to phospholipids and cholesterol through lipidic hydrogen-bonds. Free energy barriers that are associated to translational and orientational changes of melatonin have been found to be between 10–20 kJ/mol for distances of 1 nm between melatonin and the center of the membrane. Corresponding barriers for tryptophan and serotonin that are obtained from reversible work methods are of the order of 10 kJ/mol and reveal strong hydrogen bonding between such species and specific phospholipid sites. The diffusion of tryptophan and melatonin is of the order of 10${}^{-7}$ cm${}^{2}$/s for the cholesterol-free and cholesterol-rich setups.

## 1. Introduction

The cell membrane plays a central role in the control of the exchange of key elements (nutrients, wastes, drugs, and heat as the most relevant) between the exterior of a cell and its cytoplasm. Lipids, proteins, and cholesterol (CHOL) are among the main components of human cell membranes. Phospholipids are usually formed by two leaflets of amphiphilic lipids that are divided into a hydrophilic head and one or two hydrophobic tails. Such lipids can self-assemble by hydrophobicity [1,2]. Lipid bilayer membranes formed by dipalmitoylphosphatidylcholine (DPPC, $C_{40}H_{80}NO_{8}P$) and dimyristoylphosphatidylcholine (DMPC, $C_{36}H_{72}NO_{8}P$) are of great interest to computational studies [3,4,5,6] because of the abundance of experimental data [7,8,9,10,11] and their usability as model systems [12,13]. In the last decades, cell membrane systems have been extensively studied in regards to their association with drugs and small molecules [14,15,16,17,18]. Among these, we will focus our attention to three species deeply related, belonging to the indole group: melatonin (MEL, $C_{13}H_{16}N_{2}O_{2}$), its precursor serotonin (SER, $C_{10}H_{12}N_{2}O$), and its precursor tryptophan (TRP, $C_{11}H_{12}N_{2}O_{2}$), given their interest as pharmaceutical compounds with a wide variety of applications, in particular a large number that is related to sleep disorders [19,20,21]. The relationships between the three components have been reviewed a significant number of times in different contexts [22,23,24,25]. The sequence of transformations in vivo is as follows: first, the hydroxylation of L-tryptophan by tryptophan hydroxylase forms 5-hydroxytryptophan; when followed by the decarboxylation by the aromatic aminoacid decarboxylase, we get serotonin; then, if serotonin is acetylated by N-acetyltransferase to N-acetylserotonin and further methylated by 5-hydroxyindole-O-methyltransferase, we produce melatonin [22,26,27,28].

Melatonin is a neurohormone that is produced in the pineal gland, first being isolated in 1958 by Lerner et al. [29] after it was identified in bovine pineal extracts. There is evidence that MEL can regulate circadian rhythms [30] and mood, induce sleep [31], and contribute in protecting the organism from Alzheimer disease [32], having become one of most studied hormones in relation to its affectation to the human body [33,34,35,36]. MEL plays a role in aging processes, being mainly protective of oxidative stress and damage [37], and it is also related to skin pigmentation and DNA repair systems. Hence, MEL has become a very good candidate for treating several dermatoses that are associated with substantial oxidative damage, by means of the increase of intracutaneous melatonin production as well as by exogenous application and intake [37,38,39]. MEL has a significant effect on decreasing cholesterol absorption, causing a great reduction of the concentration of cholesterol in membrane bilayers and in the liver [40]. It is able to cross most physiological barriers, such as the blood-brain barrier [41,42], so that it may help to control brain function [43], and it also has interesting immunotherapeutic potential in both viral and bacterial infections [44]. MEL has been also related to the protection of the organism from carcinogenesis and neurodegenerative disorders [32,45]. Recently, the use of MEL to attenuate the effects of the severe acute respiratory syndrome (Covid-19 or SARS-CoV-2) has been under debate [46], since melatonin is a well known anti-inflammatory agent and it could be protective against viral pathogens. A comprehensive description of its functions has been summarised [47,48,49]. At the microscopical level, several works have analysed the structure and interactions of MEL with phospholipid membranes [50,51]. Both of the experiments and simulations suggest that small solutes, such as TRP and MEL, are bound to the phosphate and carbonyl regions of phospholipid species [52,53,54,55]. Recent studies indicate that cellular permeation rates in the pineal gland are of the order of 1.7 μm/s, and they can occur by pure diffusion under high temperatures and pressures [56]. However, other studies found that the active action of glucose transporters are required for the entrance of MEL inside cancer cells [57], allowing for MEL to help inhibit tumor growth [58]. The safety of MEL in humans has been addressed. Andersen et al. [59] reported that, in animal and human studies, the short term use of MEL is safe, even in extreme doses. After long-term treatments, there have only been reported mild side effects, with none of them being dangerous for human health. Experimental and computational work on mixtures of CHOL and MEL at phosphatidylcholine membranes have analysed the joint effects of the two species [60,61].

The precursors of melatonin, serotonin, and tryptophan have been thoroughly studied from long ago. SER (also known as 5-hydroxytryptamine or 5-HT) is a biogenic amine that is most noted for its role as a neurotransmitter, which is mainly produced by enterochromaffin cells in the gut and also by neurons of the brain stem [62]. It was first isolated and characterised in 1948 by Rapport et al. [63]. Serotonin was quickly identified in many tissues, including brain, lung, kidney, platelets, and in the gastrointestinal tract. It is thought to be a contributor to the regulation of human mood and happiness [64]. It has been also suggested that SER also regulates the connectivity of the brain [65]. As the third small molecule to describe here, TRP is an α-aminoacid used in the biosynthesis of proteins. Tryptophan contains an α-amino group, an α-carboxylic acid group, and a side chain indole, which makes it a nonpolar aromatic amino acid. TRP is essential in humans and it is also a precursor to the vitamin B3 and is commonly used to treat insomnia and sleep disorders, like apnea [66,67]. TRP can act as a building block in protein biosynthesis, while proteins perform a vast array of functions within organisms, such as catalysing metabolic reactions, replicating DNA, responding to stimuli, providing structure to cells and organisms, and transporting molecules from one location to another.

Our main aim here is to review and study at atomic detail the interactions and binding mechanisms between melatonin, serotonin, and tryptophan with the cell membrane, modelled as a mixture of phospholipids and cholesterol, in aqueous ionic solution. We have employed two types of computational methods, molecular dynamics (MD) and well tempered metadynamics (WTM). MD is a classical simulation tool that is able to generate a bundle of Newtonian trajectories, one for each single particle of the system, at the atomic level. As atoms interact through pairwise force fields, their trajectories (composed of positions and linear momenta of all particles) are deployed and stored at regular time intervals, in order to be analysed using tools from Statistical Mechanics [68]. MD is a versatile method that is able to successfully reproduce a large number of microscopic properties of a wide variety of systems, from simple atomic liquids, such as argon [69] to molecular liquids as water [70,71], aqueous solutions at interfaces [72,73,74,75,76], up to complex biophysical systems like DNA [77,78,79] or model cell membranes [6,80,81,82,83,84]. In order to handle the problem of computing free energy landscapes in multidimensional systems, different classes of methods have been proposed, such as quantum mechanics/molecular mechanics [85], transition path sampling [86,87,88,89,90,91,92], adaptive biasing force [93], umbrella sampling methods [94,95], density functional theory molecular dynamics [96], or calculations of potentials of mean force [97] based on reversible work methods [98]. In this work, we have employed reversible work and WTM, a method that is able to efficiently explore free energy surfaces of complex systems while using multiple reaction coordinates what has been revealed to be very successful [99] for a wide variety of complex systems [100,101,102,103,104]. Section 3 reports the technical characteristics of all simulations.

## 2. Results and Discussion

### 2.1. Structural Properties of the Membranes

The structural characteristics of the membranes and the local distributions of atomic species are the first group of properties to be analysed. To do so, we have sketched the detailed atomic structures of the three small molecules, the two phospholipids composing the membranes (DMPC, DPPC) and CHOL in Figure 1. There, the highlighted sites of TRP are the zwitterions ‘H1’, sharing a positive charge between the three hydrogens that are bound to ‘N1’; ‘H2’, bound to ‘N2’ and the zwitterions ‘O1’ and ‘O2’, bound to ‘C1’ and sharing a negative charge. In SER, we highlight ‘H1’, ‘H4’, and ‘O’, and, for MEL, we will keep special attention into ‘O1’, ‘O2’, ‘H15’, and ‘H16’. Finally, the sites ‘O1’ and ‘O2’ sharing the negative charge and oxygen atoms ‘O6’, ‘O8’ will be considered for DPPC and DMPC and the hydroxilic OH pair for CHOL.

A common test in computer simulations of cell membranes is the comparison of the area per lipid and thickness of the membrane with experimental data from scattering density profiles [105]. We have monitored the surface area per lipid *A* when considering the total surface along the XY plane (plane parallel to the bilayer surface) that is divided by the number of lipids $N_{l}$ plus the number of cholesterol molecules $N_{chol.}$ in one lamellar layer [106], as defined in Equation (Equation 1):(1)$$A=\frac{L_{x}\times L_{y}}{N_{l}+N_{chol}},$$
where $L_{x}$ and $L_{y}$ are the length of the simulation box along *X*-axis and *Y*-axis, respectively. *Z*-axis is the (instantaneous) normal direction to the surface of the bilayer, set along plane XY. Fluctuations in the thickness of the membrane are related to the effect of cholesterol on the rigidity of the membrane and its capability to allow the passing of species in and out of the cell. In this work, we defined the thickness Δz as the distance between the phosphate groups of the lipids at the two sides of the membrane. The area per lipid and thickness along the last 500 ns of each simulation have been computed (see Figure 2) and Table 1 reports the average values.

The area per lipid decreases as cholesterol concentration increases: this is a well known trend, as will see below. We obtained a value of around 0.61 nm${}^{2}$ for a cholesterol-free system and smaller values down to 0.40–0.42 nm${}^{2}$ when cholesterol has been incorporated. In all cases, the area per lipid is practically independent of the small molecule that is imbedded in the membrane, and it has little influence of the main type of phospholipid. These results are in excellent agreement with experimental data [109,110], where the value for pure DMPC is of about 0.6 nm${}^{2}$ at 303 K. Further, and according to Nagle et al. [1], values of *A* of pure DMPC membranes can be obtained from multiple methods (neutron scattering, X-ray and NMR) and they have been reported to be between 0.59 and 0.62 nm${}^{2}$ at the liquid crystal phase. In the case of DPPC, the best estimations were of between 0.48 and 0.52 nm${}^{2}$ in the gel phase (293 K) and 0.64 nm${}^{2}$ in the liquid phase. These results are also in overall good agreement with other computational data in a wide variety of thermodynamical conditions [11,109,111,112,113], where the values for pure DPPC ranged between 0.50 and 0.63 nm${}^{2}$ and the trend of decreasing areas for increasing cholesterol percentages was clearly reported. The huge change that is produced at 30% cholesterol concentration is consistent with the fact that phosphatidylcholine membranes experience a phase transition liquid disordered (cholesterol-free system) to liquid ordered phase (systems of cholesterol 30% and 50%) [114,115].

The thickness of the membranes are in good agreement with those that were reported by Kucerka et al. [110] by means of X-ray and neutron scattering. The reported value was of 3.67 nm at 303 K for the DMPC membrane at 0% cholesterol. From the results that are reported in Table 1, we obtain values around 3.5–4 nm for pure bilayers and of 4.4–4.9 nm when cholesterol is considered. We observe a tendency to larger bilayer thickness for increasing cholesterol concentration. Given the reduction of the area per lipid at high cholesterol percentages, we can conclude that cholesterol favours the compression of the structure of the bilayer membrane. This feature can increase the rigidity of the membrane and, by extending the tails of the lipids, give larger bilayer thickness. Such an increase of the rigidity of the membrane was observed from both experimental and computational sides [60,61] in MEL-CHOL mixtures nearby phosphatidylcholine bilayers. According to these studies, the effect of MEL reducing the thickness of the membrane and enhancing its fluidity was partially compensated by the condensating effect of cholesterol.

### 2.2. Preferential Localisations of the Small Molecules at the Interfaces of Phospholipid Membranes: Atomic Radial Distribution Functions

Each of the three small molecules considered in the present work has been simulated for long MD trajectories of hundreds of nanoseconds. We have monitored their positions and velocities and obtained structural, energetic, and dynamical information. In this section, we will focus our attention on the local structure of the probes when embedded in the membrane. As a general fact, we have observed that all three selected species show a strong tendency to be continuously adsorbed at the interface of the membrane during long periods of the order of 10 ns. In the remaining time, the small molecules move away to be solvated by the ionic solution surrounding the membrane. As an example, in Figure 3 we report the evolution in time (window of 60 ns) of the position of the center of mass of melatonin when adsorbed at a DMPC-cholesterol membrane.

In Figure 3, we can observe that the influence of cholesterol is of paramount importance: when the concentration of cholesterol in the membrane reaches 50% of all lipids, MEL can easily shift between the interface of the membrane and the solvating aqueous ionic solution, but, at lower concentrations, the small molecule is likely inside the membrane during the whole time span considered. This indicates that moderate changes of cholesterol concentration may induce some specific organic probes to retreat from the inside of cellular membranes to the outer regions and remain outside the cell. This might have strong implications in melatonin delivery. The relationships between MEL and CHOL and their interactions have been studied since long time ago [40,60,61], but the knowledge of their effects are still quite elusive.

Because the computation of radial distribution functions (RDF) is the best way to investigate atom–atom local structures, we have computed a series of specific RDF in order to have an overview for TRP and MEL. We define the RDF for an atomic pair composed by particles ‘1’ and ‘2’ as $g_{12}\left(r\right)$, and it is given by:(2)$$g_{12}\left(r\right)=\frac{V\;\langle n_{2}\left(r\right)\rangle }{4\;N_{2}\;\pi r^{2}\;\Delta r},$$
where $n_{2}\left(r\right)$ is the number of atoms of species ‘2’ surrounding a given atom of species ‘1’ inside a spherical shell of width Δr. *V* stands for the total volume and $N_{2}$ is the total number of particles of species ‘2’. In the case of TRP, we have considered the partial RDF that is reported in Figure 4, whereas, for MEL, we will analyse the RDF that is presented in Figure 5.

From the data that are reported in Figure 4, we can observe that TRP stays bound to the inner part of the membrane during long periods of time, according to the time scale of our simulations, in good agreement with the results indicating that, in a cholesterol-free DOPC bilayer membrane, TRP is preferentially located in the interfacial region [55]. In this work, we have observed that the average continuous lifetime of TRP at the interface of the DPPC bilayer is of the order of 10 ns (data not shown). From Figure 4, hydrogen bond (HB) connections between sites ‘H1’ and ‘H2’ of TRP and DPPC sites ‘O2’ and ‘O8’ (labels according Figure 1) have been found. HB are very short, since the maxima of the g(r) related to ‘H1’ hydrogens in TRP are located around 1.7 Å for ‘O2’ and around 1.75 Å for ‘O8’ of DPPC. Accordingly, the presence of cholesterol reduces ‘H1-O2’ binding, but enhances the ‘H1-O8’ one. As a general fact, the presence of cholesterol increases the length of HB, but also making such bonds stronger. This indicates that the influence of the cholesterol in the TRP-DPPC binding is a major effect. Interestingly, ‘H2’-DPPC binding was observed in all of the analysed setups, but with maxima found at larger distances (1.9–2.0 Å). Again, the presence of cholesterol showed a major influence on the characteristics of hydrogen bonding.

Figure 5 reports the structural results for MEL. All RDF show fluctuating profiles, especially at distances that are larger than *r* = 3 Å and beyond (higher order coordination shells). There is a clear first coordination shell in all cases, located around 1.8–2.0 Å, due to HB between MEL and the remaining species, such as in the TRP case. The largest maximum of all RDF is the one for MEL-CHOL association (not shown here), which is centered at 1.9 Å when the concentration of cholesterol is of 30%. Choi et al. reported interactions of MEL-CHOL in DPPC bilayers [61], but at finite melatonin concentration. In the remaining cases, HB lengths are around 1.9 Å and they were between both ‘H15’ and ‘H16’ of MEL and DMPC sites ‘O1’ (or ‘O2’, since both of the sites are sharing the negative charge of the zwitterion). In a similar fashion, MEL can also form HB between both ‘H15’ and ‘H16’ with the DMPC’s sites ‘O6’ (or ‘O8’) for all three percentages of cholesterol. The present findings are in good agreement with the experimental data from Severcan et al. [50] that were obtained by Fourier transform infrared spectroscopy, who observed hydrogen bonding connections between the N-H group of the furanose ring of MEL (‘H16’ in this work) and the carbonyl (C=O) and phosphate (PO${}_{4}$) groups in DPPC membranes. Our results indicate HB between ‘H15’ and ‘H16’ of MEL with DMPC’s phosphate group (‘O1’) as well as with the more internal carbonyl groups (‘O6’ and ‘O8’). The previously unobserved hydrogen bonds of ‘H15’ with the two well known acceptor groups in the phosphatidylcholines indicated above are responsible for the absorption of MEL into the membrane deeper than TRP, with the two selected donors (‘H15’ and ‘H16’), together with the ‘H15-O Chol.’ bridges. These findings are in excellent agreement with the results that were reported by Drolle et al. [60] by means of small angle neutron diffraction and MD simulations.

### 2.3. Orientational Distributions of Melatonin

Several previous studies have shown that the orientations of drugs on membranes significantly impact their function in cells [116,117,118,119,120]. In the present work, we have computed the principal orientations of MEL through the definition of three different dihedral angles. We have observed that, in all cases, two preferential orientations arise, since the averaged angular distributions of MEL are centered around two well defined angular values, which we call “folded” and “extended” configurations of MEL, found at all cholesterol concentrations. The dihedral angle, which has a better distribution regarding its fluctuations around mean values, is the angle Ψ that is represented in Figure 6.

The torsional angle considered here is related to the nitrogen atom labelled ‘N1’ in Figure 1, namely the nitrogen chemically bound to the hydrogen labelled ‘H15’. For this angle Ψ, after analysing 100 ns of equilibrated trajectories (production runs), we found averaged values that corresponded to 81 ± 10°, (folded) and 170 ± 23°, (extended), as shown in Figure 7. Further, from the distributions reported there we can observe that Ψ is neatly defined and it reaches nearly the same mean value, regardless of the concentration of cholesterol of the system. Interestingly, we find that the extended configuration of MEL is most favoured in the case of the highest concentration 50% (green triangles), which suggests that introducing cholesterol into the system could help MEL change from its folded to its extended configuration more easily through hydrogen-bonding between MEL-DMPC and MEL-cholesterol. In addition, according to this, Ψ is an excellent candidate for being used as a collective variable in metadynamics calculations [121,122] of free energy landscapes for MEL binding in biomembranes, as we report in Section 2.4.

### 2.4. Free Energy Profiles of Small Molecules and Free Energy Hypersurfaces of Melatonin Binding

Once we have established preferential locations and angular distributions of the small molecules, if assuming some of these coordinates as good candidates for collective variables, we are ready to use the WTM technique to obtain precise, quantitative values of the free energy barriers that need to be surmounted by the small molecules to move throughout the system, mainly exchanging positions between the interfacial regions and the bulk like aqueous regions of the system. Computationally speaking, WTM is a very expensive method that requires very long trajectories, so that the target subsystem, i.e., the small molecule, can move in the full configurational space, visiting regions of low energy with high probability as well as regions of high energy, being very unlikely to be accessed. In this work, we will complement WTM with a much simpler technique, based in the knowledge of RDF described above, namely the computation of the reversible work that is needed for the target to move between selected regions, being indexed by one dimensional coordinate, such as a radial distance *r*. The theory has been nicely described in chapter 7 of Ref. [123]. It states that we can obtain $W_{12}\left(r\right)$ i.e., the reversible work (sometimes also known as potential of mean force, PMF) that is required to move two tagged particles from infinite separation to a relative separation *r* from:(3)$$W_{12}\left(r\right)=-\frac{1}{\beta }\ln g_{12}\left(r\right),$$
where $\beta =1/(k_{B}T)$ is the Boltzmann factor, $k_{B}$ the Boltzmann constant and *T* is the temperature. $W_{12}\left(r\right)$ can be understood as the relative Helmholtz (canonical ensemble) or Gibbs (isothermal-isobaric ensemble) free energy that is associated to atomic pairing. Figure 8 reports the W(r) found for the three small molecules and Table 2 reports the quantitative estimations of the main energy barriers.

Reversible work calculations can only give us a rough approach to the size of real barriers, since it is based in the use of the interparticle distance *r* as the only reaction coordinate, which is known to produce some underestimation [97]. However, given that accurate reaction coordinates are usually unknown, very hard to obtain, and multidimensional, W(r) is a reasonable way to estimate the order of magnitude of the free energy barriers. The sata reported in Table 2 and Figure 8 reveal to us that the highest barrier corresponds to the pairing of ‘H1’ of TRP with ‘O2’ of DPPC. We have found that all small molecules are able to establish HB with ‘O2’ and also with the site ‘O8’ of DPPC, with the latter being located deeper in the membrane (see Figure 1). Conversely, we did not find bindings between ‘H15’ site of MEL and ‘O2’ site of DPPC. The position of maxima of the first barrier are mostly centered around 2.45 Å for small molecule-‘O2’ binding, whereas barriers of ligand ‘H4’ of serotonin that are associated to the ‘O8’ sites are centered around 2.75 Å. In the case of SER, only a first minimum is clearly found, which indicates that SER is normally bound to the plasma membrane and it does not move to the solvent bulk.

The binding of ‘O2’ in DPPC to TRP is located at 1.75 Å, corresponding to the first minimum of the PMF between TRP and DPPC, which is of the order of the typical HB distance in water. Nevertheless, stable positions for ‘O8’ sites of DPPC are found between 1.7 and 2 Å, a remarkable wider distance. We can compare these values with the barrier for TRP (attached to a polyleucine α-helix) inside a DPPC membrane of 12 kJ/mol [52] or the barrier of the order of 16 kJ/mol found for TRP in a dioleoylphosphatidylcholine bilayer membrane [55]. Finally, the agreement of the barriers reported in the present work (Table 2) with other neurotransmitters, such as glycine, acetylcholine, or glutamate, of around 2–5 kJ/mol when it is located close to the lipid glycerol backbone [124], is also quite remarkable.

One way of getting much more precise free energy estimations is through methods operating with multidimensional reaction coordinates. One of best methods is well tempered metadynamics, although it is a very expensive computational tool, as we will explain in Section 3.2. As a specific example, we have applied WTM to the calculation of the hypersurface of free energy for the system that is composed by MEL and DMPC, at the three cholesterol concentrations of 0%, 30%, and 50% that are described in Section 3.1. The WTM specifications have been reported with full details in Section 3.2. We need to define several specific collective variables (CV) that are able to meaningfully describe characteristic configurations of MEL in order to compute the three sets of two dimensional (2D) well tempered metadynamics simulations. The results shown Figure A2 give us an indication of the convergence of WTM. To achieve this goal, we had to run trajectories of 1400 ns. These trajectories followed from the MD production runs that were employed to obtain structural and dynamical information.

Figure 9 shows the resulting 2D free energy surfaces (FES) of MEL bound to DMPC membranes and they correspond to Gibbs free energy calculations. Each state has been indexed by two CV: (1) the *z* distance between the center of mass of MEL and the center of the membrane (z=0); (2) the torsional angle Ψ defined and analysed in Section 2.3. The inspection of Figure 9 shows that regions with clear minima are present in the FES in all cases. The main features are the global minima of the FES located between z∈[0.7,3] nm and around two distinctive values for the dihedral angle, namely those that are around |Ψ|∼[70°,180°]. Such orientations are in excellent agreement with the average values of Ψ that were obtained from ordinary MD simulations (see Figure 7) that correspond to the two “folded” and “extended” geometries of melatonin previously reported.

The 2D free energy landscapes reveal that the most favourable stable states of melatonin binding to the membrane (basins A,B,C,D) correspond to *z*-distances around 0.8 nm at the cholesterol-free system, whereas such a distance tends to significantly increase around to 1.3 nm for the 3% cholesterol concentration and up to 2.3 nm when cholesterol reaches 50%. As a general fact, the 2D surfaces that are shown in Figure 9 correspond to contour plots with values being referred to a global zero. The zero of each 2D plot has been set at the highest free energy value of all, in our case corresponding to locations at the computed maxima of the coordinate *z*.

According to the CV1, MEL is preferentially located at the interface of the DMPC-cholesterol bilayer (regions with 0.8<z<3.0 nm). The locations of MEL outside the interface and far enough of lipid headgroups (z>4.3 nm) show very larger free energies and they cannot be considered to be stable states of the system. Those regions will be considered as the “bulk”, i.e., the region containing the electrolyte solution surrounding the membrane. When considering the information revealed by CV2, we can distinguish two sets of minima: (1) for |Ψ|=67° (basins B and C) and (2) for |Ψ|=180° (basins A and D. These minima are related to the two preferential configurations of MEL close to a DMPC-cholesterol bilayer (folded, extended) indicated above around 80 and 170° (see Section 2.3).

We collected the data extracted from Figure 9 to estimate the main free energy barriers for the main configurational changes on MEL in the quantitative side. Table 3 reports the values.

Our findings have revealed a rather wide range of absolute free energies, which are in good agreement with the range that was reported by Jämbeck and Lyubartsev [125] for small molecules (ibuprofen, aspirin, and diclofenac) at the surroundings of lipid bilayers, of the order of free energy ranges up to 70 kJ/mol and barriers around 40 kJ/mol. For the sake of comparison, we should remark that the barriers of 2–10 kJ/mol reported in Table 2 obtained from the PMF of Figure 8 were related to the formation and breaking of HB, when the small molecules were located inside the interfacial region, regardless of its orientation. However, free energy barriers of orientational changes or those that are related to large displacements of MEL to the center of the membrane or to the extracellular bulk are much larger. For instance (see Table 3), the free energy that is required to exchange between folded and extended MEL configurations is very stable, around 15–20 kJ/mol for all cholesterol concentrations. Florio et al. [126] using a combination of several fluorescence and spectroscopic techniques, found the conformational preferences of an isolated MEL molecule under molecular beams. These authors found MEL three *trans* and two *cis* conformers showing free energy gaps of approximately 12.5 kJ/mol, in quantitative agreement with the values reported here for orientational changes.

However, the barrier that is to be surmounted by MEL to move from the interface of the membrane to the extracellular fluid is strongly dependent on cholesterol concentration. We observed that it decreases with larger amounts of cholesterol, between 25 kJ/mol at the cholesterol-free case to around 10 kJ/mol for the 50% concentration. Finally, the probability for MEL to access the central, hydrophobic regions of the membrane is scarce, since it will require surpassing free energy barriers of more than 40 kJ/mol. This will make it very difficult to observe transmembrane crossings in the simulated scale of 1 μs. In a recent work conducted by Wang and coworkers [127], small solutes, such as glycerol, caffeine, isopropanol, or ethosuximide, were simulated nearby a model cell membrane. These authors found that, in order to observe transmembrane crossings of such small solutes in the time length of a simulation at the atomic level of description, they needed to run trajectories of 10 μs at low temperatures (310 to 330 K) or, alternatively, raise the temperatures to more than 400 K (for simulation times of 1 μs). In our case, we did not record any transition of MEL between the two sides of the DMPC membrane along the simulated trajectory.

### 2.5. Diffusion Coefficients of Small Molecules: Tryptophan and Melatonin

Microscopic translational dynamics of tryptophan and melatonin have been considered. We have evaluated the mean square displacement (MSD) of the carbon ‘C2’ in TRP (see Figure 1) and of the center of mass of MEL. From the long time slopes of both MSD, we obtained the corresponding self diffusion coefficients *D* through the Einstein formula of Brownian motion:(4)$$D=\lim_{t\to \infty }\frac{<{\left|{\vec{r}}_{i}\left(t\right)-{\vec{r}}_{i}\left(0\right)\right|}^{2}>}{2\;d\;\Delta t},$$
where ${\vec{r}}_{i}\left(t\right)$ is the instantaneous position of particle *i*. In this general procedure, the spatial dimension of the diffusion regions *d* is considered. TRP and MEL showed lateral like diffusion (d=2). Table 4 summarises the results.

The main finding is that self-diffusion coefficients *D* for TRP (Table 4) are between 3–14 $\times \;10^{-7}$ cm${}^{2}$/s, which, even within the same order of magnitude, are significantly larger than those of the diffusion of DMPC molecules [6] (0.6 $\times \;10^{-7}$ in the absence of cholesterol). The main trend is that *D* increases for rising cholesterol concentrations. Overall, we find that the mobility of TRP is significantly higher than that of DMPC. Nevertheless, the effect of temperature is remarkable here, since, at complementary simulations at 310 K, TRP diffusion was of about 2 $\times \;10^{-7}$, i.e., being significantly slower given the gel like state of the membrane in such a case.

In the case of MEL, *D* also shows a tendency to increase when cholesterol is mixed with DMPC, regardless of its concentration. In Table 4, at 30% cholesterol, the value of *D* for MEL is six times larger than the value of *D* of DMPC molecules in pure DMPC bilayer membrane systems [6]. This fact would suggest that its mechanisms of diffusion may be similar to those of an individual particle (such as in Fickian diffusion) and qualitatively different of those of lipids, whose diffusion was observed to occur in a sort of collective way, being associated in local groups of a few units (around 5–10 units) [6].

## 3. Methods

### 3.1. Molecular Dynamics

We have performed seven independent series of MD simulations for TRP, SER, and MEL in different environments (DMPC, DPPC, and different concentrations of CHOL, namely: 0%, 30%, and 50% for TRP and MEL, whereas, for SER, only the cholesterol-free membrane was simulated. Each system contains a total of 204 lipid and/or cholesterol molecules that were fully solvated by ∼5000–10,000 TIP3P water molecules and 17–21 sodium chloride pairs at the human body concentration (0.15 M), yielding a system size of about 40,000–60,000 atoms. Table 5 sumarises the characteristics of all simulations.

Our MD inputs were created with the CHARMM-GUI web-based tool [128]. All of the systems were simulated at the the isobaric-isothermal ensemble. i.e., at constant number of particles (N), pressure (P), and temperature (T) conditions, with equilibration periods for all simulations being more than 200 ns. After equilibration, we recorded statistically meaningful trajectories of more than 600 ns. A typical size of the system was of 80Å × 80Å × 81Å. The simulation time step was of 2 fs in all cases. Given its ability to reproduce area per lipid of DMPC and DPPC in excellent agreement with experimental data, the CHARMM36 force field [129,130] was used. All of the bonds involving hydrogens were fixed to constant length, allowing for fluctuations of bond distances and all sorts of angles for the remaining atoms. Van der Waals interactions were cut off at 12 Å with a smooth switching function starting at 10 Å. Long ranged electrostatic forces were taken into account by means of the particle mesh Ewald method [131], with a grid space of about 1 Å and updated every time step. The periodic boundary conditions were considered in each spatial direction.

### 3.2. Well Tempered Metadynamics

As we pointed out above, obtaining free energy profiles and estimating the height of the main barriers between stable states is a very difficult task in condensed matter systems [132]. In the present work, in Section 2.4 we have presented two possible pathways to do the job: (1) using a direct method that is based on the reversible work theorem, but knowing that it is, at its best, a first approach to the real barriers and (2) employing a more sophisticated tool, called “metadynamics”, which, given a well chosen set of a few reaction coordinates (the collective variables), is able to provide a much more exact picture of the free energy hypersurface. Huber et al. [133] and Grubmüller [134] initially proposed the method and it was developed later on by Laio and Parrinello [99,121] as a method to explore multidimensional free energy surfaces as a function of a *a priori* unknown CV. Given some deficiencies of the original method, well tempered metadynamics [122,135] was introduced. In the present work, we have run 1.4 μs well-tempered metadynamics simulations in order to obtain Gibbs free energies of the binding states of MEL at phospholipid membrane surfaces that were made by DMPC lipids and CHOL in sodium chloride aqueous solution. Starting from the long trajectories generated by unbiased MD simulations for MEL-DMPC, we could make a reliable guess of two potentially appropriate CV. All of the metadynamics simulations were carried out by means of the PLUMED2 plugin [136,137] within the joint GROMACS/2018.3-plumed tool and they were performed in the NPT ensemble. Table 6 reports the particular details of the WTM simulations. The ussual periodic boundary conditions in all directions of space were considered.

## 4. Conclusions

The interactions of some small molecules with human cells are undoubtedly a relevant field of research. In particular, the hormone melatonin has an important role in the treatment of a wide variety of diseases and problems that are related to sleep. It works as a regulator of circadian rhythms and as an antioxidative. Further, its precursor serotonin is a neurotransmitter playing a key role in a variety of physiological processes and in the regulation of mood and cognitive learning. Serotonin is synthesised by the body from its precursor, the essential aminoacid tryptophan. Tryptophan is a zwitterion, with a protonated amino group (NH${}^{3+}$) and a deprotonated carboxylic acid (COO${}^{-}$), and it is used as an antidepressant. In the present work, we are reviewing a series of MD and WTM simulations of different lipid bilayer membranes in an aqueous ionic solution of NaCl with embedded small molecules. The calculations have been performed using the CHARMM36 force field. Among them, cholesterol at two concentrations (30% and 50%) has been considered together with the cholesterol-free reference systems in order to explore the influence of CHOL concentrations on the properties of the small molecules.

In a preliminary study on the adsorption of tryptophan at a DPPC bilayer membrane at 310.15 K (gel phase) [138], we observed a strong first coordination shell for TRP-water and TRP-DPPC pairs. In this study, we focussed in the liquid phase and only found relevant changes in the local structure and dynamics of TRP for cholesterol concentrations above 30%. TRP-DPPC binding involved coordination shells for the different oxygen sites of DPPC that are able to associate (‘O2’ and ‘O8’) versus the two tagged hydrogens (‘H1’ and ‘H2’) in TRP. Additionally, the distribution functions of TRP-CHOL revealed very stable hydrogen bonding. TRP is able to establish strong interactions with all solvating particles (water, DPPC, and CHOL), including a sort of double bridge between DPPC and cholesterol species. Typical HB distances have been found to be around 1.7–2.0 Å, which is in good agreement with experimental data [139]. Finally, the self diffusion coefficients of TRP are of the order of 10${}^{-7}$ cm${}^{2}$/s, being strongly dependent of cholesterol’s concentration.

In the case of melatonin, we have simulated its behavior when embedded in a cholesterol rich DMPC membrane at 303 K and 1 atm. Our interest was firstly focused on the local structure and angular distributions of MEL. In a similar fashion as in the case of TRP, strong hydrogen bonds between MEL-DMPC and MEL-CHOL have been found. The most important structures of MEL have been observed for two angular configurations: “folded” and “extended”. Using a particular dihedral angle (Ψ), we observed two preferential values. The angle Ψ was revealed to potentially act as a reliable reaction coordinate, since two neat angular distributions around ∼81° and ∼170° were clearly distinguished. We also observed that introducing cholesterol into the system can favour MEL to exchange between extended and folded configurations. Again, the self diffusion coefficient of MEL was found to be of the order of 10${}^{-7}$ cm${}^{2}$/s, although, in this case, with a very mild dependence on cholesterol’s concentration.

Free energy barriers for serotonin, melatonin, and tryptophan at 323.15 K and 1 atm have been analysed using the reversible work theorem, which provides us a simple way to estimate the height of the barriers that are related to the interatomic distances. These features will be directly related to the formation and breaking of hydrogen-bonds. We have found marked first and second coordination shells that correspond to two minima of the PMF, with energy barriers for TRP-DPPC of the order of 10 kJ/mol. Most remarkable have been the binding between hydrogen ‘H1’ of TRP and oxygens ‘O2’ and ‘O8’ of DPPC. In the case of serotonin, we have found it to be a molecule strongly anchored at the membrane unlike to be solvated by water. Interestingly, melatonin has revealed to be able to interact both with water and DPPC, still showing moderately strong free energy barriers. In order to get more precise information, we have conducted well-tempered metadynamics simulations, and obtained 2D free energy landscapes for MEL binding to the DMPC-CHOL membranes. Two CVs have been considered: a dihedral angle Ψ and the distance *z* between the center of mass of MEL and the center of the lipid bilayer (set at z=0). From our results, we have found that MEL is usually bound to the external side of the membrane, at distances z∼1–2 nm and in two main configurations with Ψ = 70° (folded) and 180° (extended), with an energetic cost for the exchange between the two conformations of about 15–20 kJ/mol. After CHOL is introduced into the system, it pushes MEL to escape outside the interfacial region of the membrane and move away until it is fully solvated by the aqueous ionic solution. The energetic cost for MEL to leave the interface of the membrane towards the water bulk (*z*-distances around 4 nm) has been estimated at ∼10–25 kJ/mol. A very uncommon situation in our simulations was that of MEL accessing the center of the membrane, an energetically expensive process (free energy barriers of 40 kJ/mol). We believe that the findings that are presented in this work could be of practical use in two ways: (1) for the design of new reaction coordinates in similar systems of small molecules of biochemical interest, such as amino acids, neurotransmitters, drugs, or hormones and (2) from a more general perspective, to contribute the unveiling of the microscopic interactions of small molecules with cell membranes and the key role that is played by cholesterol in the properties of such molecules. All of this can lead to advances in the research of new pharmaceutical compounds and to a better understanding of the currently available ones.

## Figures and Tables

**Figure 1 ijms-22-02842-f001:**
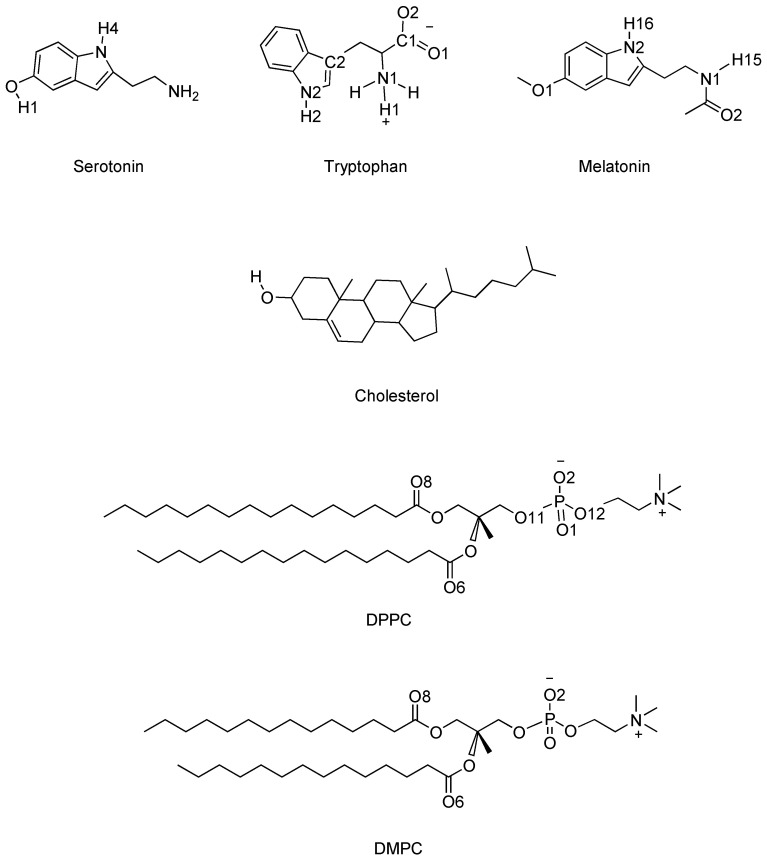
Atomic structures of melatonin, serotonin, tryptophan, dipalmitoylphosphatidylcholine (DPPC), dimyristoylphosphatidylcholine (DMPC), and cholesterol. Backbone hydrogens are not explicitly shown. The highlighted labels will be referred in the text.

**Figure 2 ijms-22-02842-f002:**
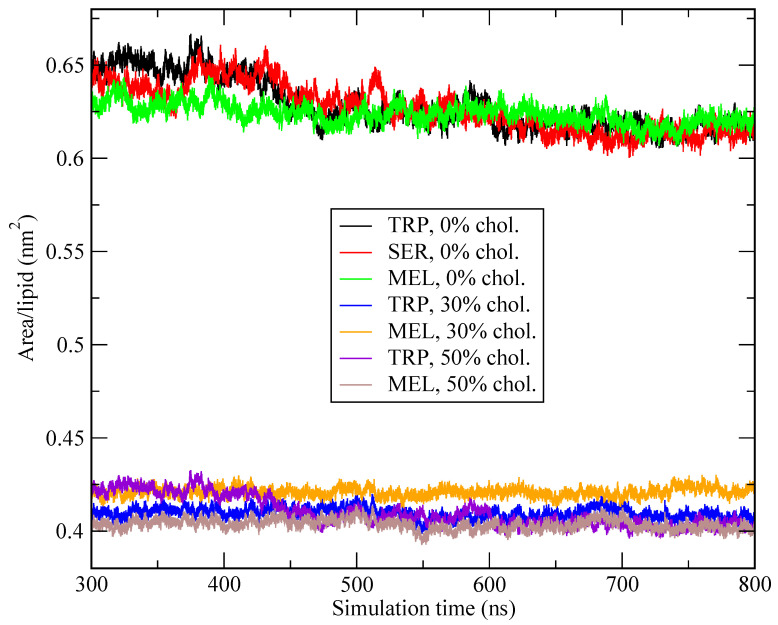
Area per lipid of systems with different cholesterol contents: 0%, 30%, and 50% as a function of simulation time.

**Figure 3 ijms-22-02842-f003:**
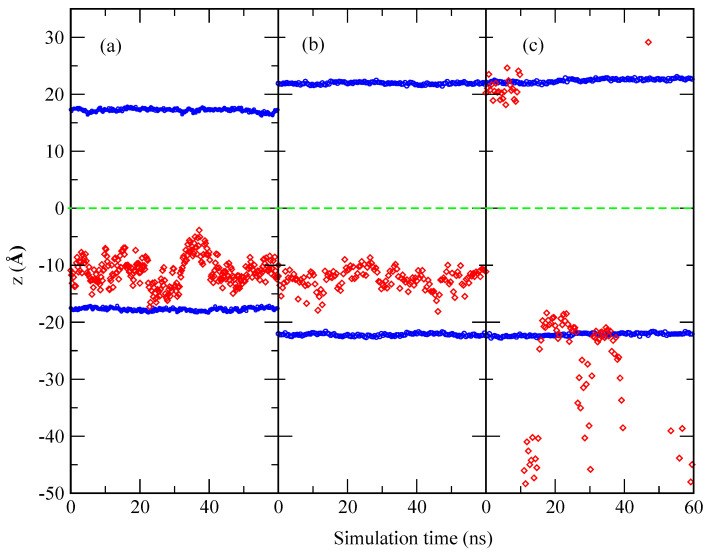
*Z*-axis location of the center of mass of MEL in a DMPC lipid membrane with different cholesterol contents as a function of simulation time. The green dashed line indicates the geometrical center of the bilayer membrane. Data partially taken from Ref. [108]. (**a**) 0% cholesterol, (**b**) 30% cholesterol and (**c**) 50% cholesterol. Red diamonds indicate the position of MEL and blue circles indicate the position of phosphorous atoms of DMPC lipids.

**Figure 4 ijms-22-02842-f004:**
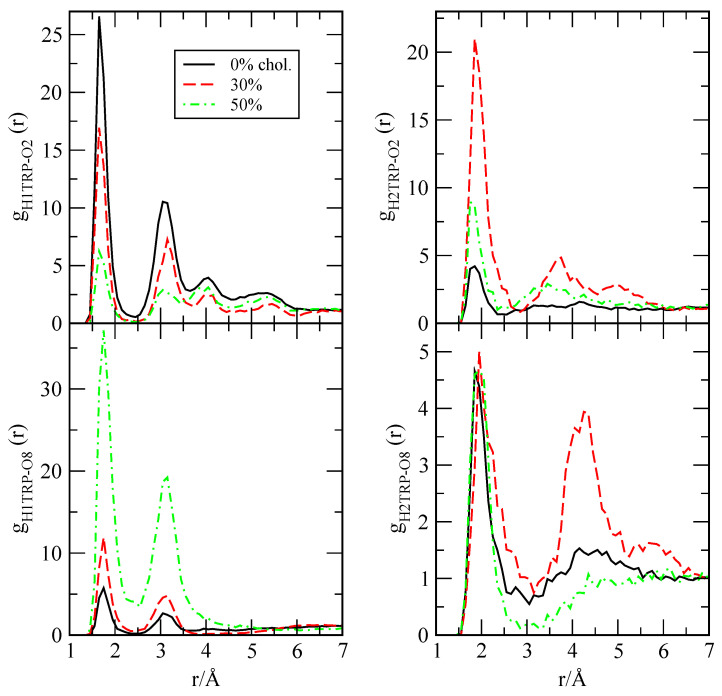
Radial distribution functions for TRP with DPPC (charged sites ‘H1’, ‘H2’, ‘O2’, and ‘O8’, see Figure 1): H1TRP-O2 (**top left**), H1TRP-O8 (**bottom left**), H2TRP-O2 (**top right**), and H2TRP-O8 (**bottom right**). Data are partially taken from Ref. [107].

**Figure 5 ijms-22-02842-f005:**
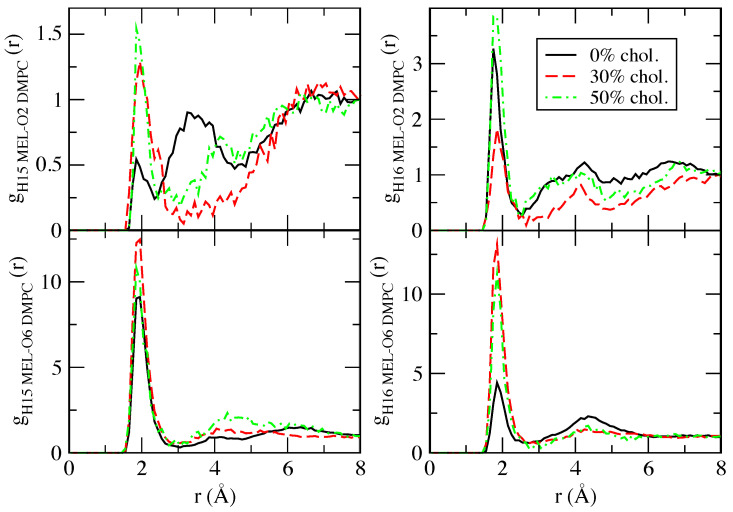
Selected radial distribution functions for hydrogens of MEL (‘H15’ and ‘H16’) with DMPC (‘O2’ (representing ‘O1&O2’) and ‘O6’ (representing ‘O6’ and ‘O8’). The labels as in Figure 1. Data are partially taken from Ref. [108].

**Figure 6 ijms-22-02842-f006:**
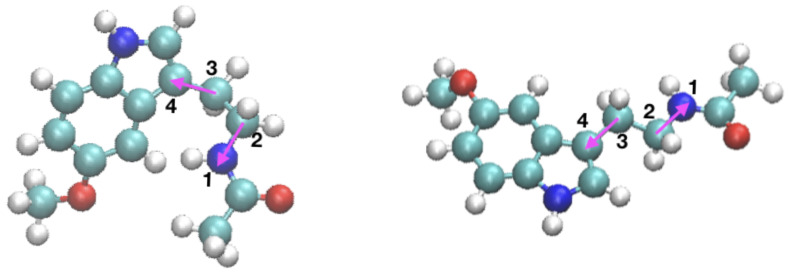
Two principal configurations of MEL: folded (**left**) and extended (**right**), indicated by the dihedral (torsional) angle Ψ. The atoms forming the melatonin molecule are: carbon (cyan), oxygen (red), hydrogen (white), and nitrogen (blue).

**Figure 7 ijms-22-02842-f007:**
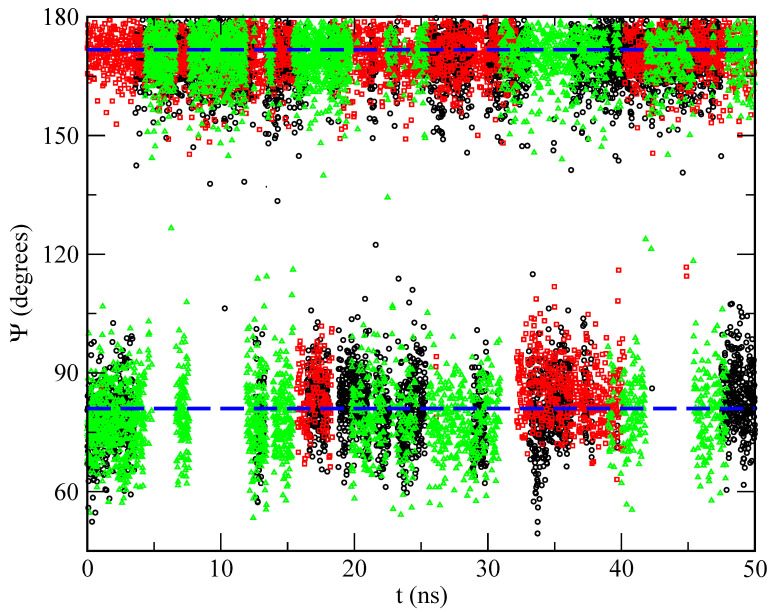
Angular distributions for the selected dihedral angle Ψ as a function of simulation time. Percentages of cholesterol are: 0% (black circles), 30% (red squares), and 50% (green triangles). The dashed lines indicate average values and are a guide for the eye.

**Figure 8 ijms-22-02842-f008:**
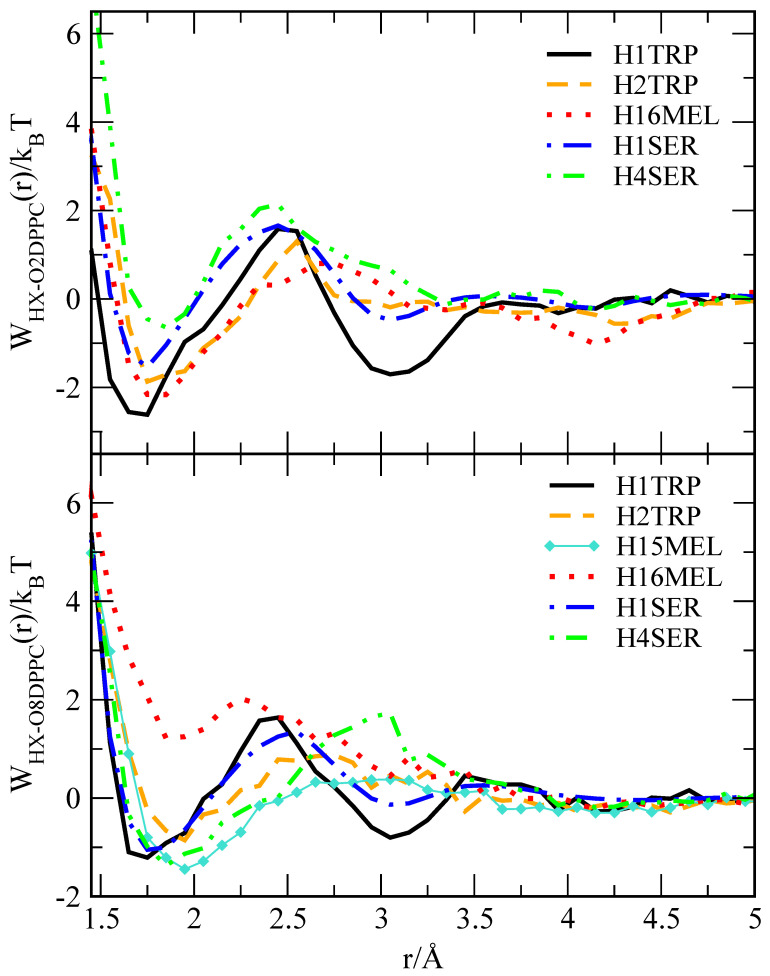
Reversible work W(r) (in $k_{B}T$) for DPPC-small molecules: TRP, SER and MEL. In the present system 1 $k_{B}T$∼2.7 kJ/mol. Hydrogen and oxygens of DPPC are indicated with same labels, as described in Figure 1. Data are partially taken from Ref. [98].

**Figure 9 ijms-22-02842-f009:**
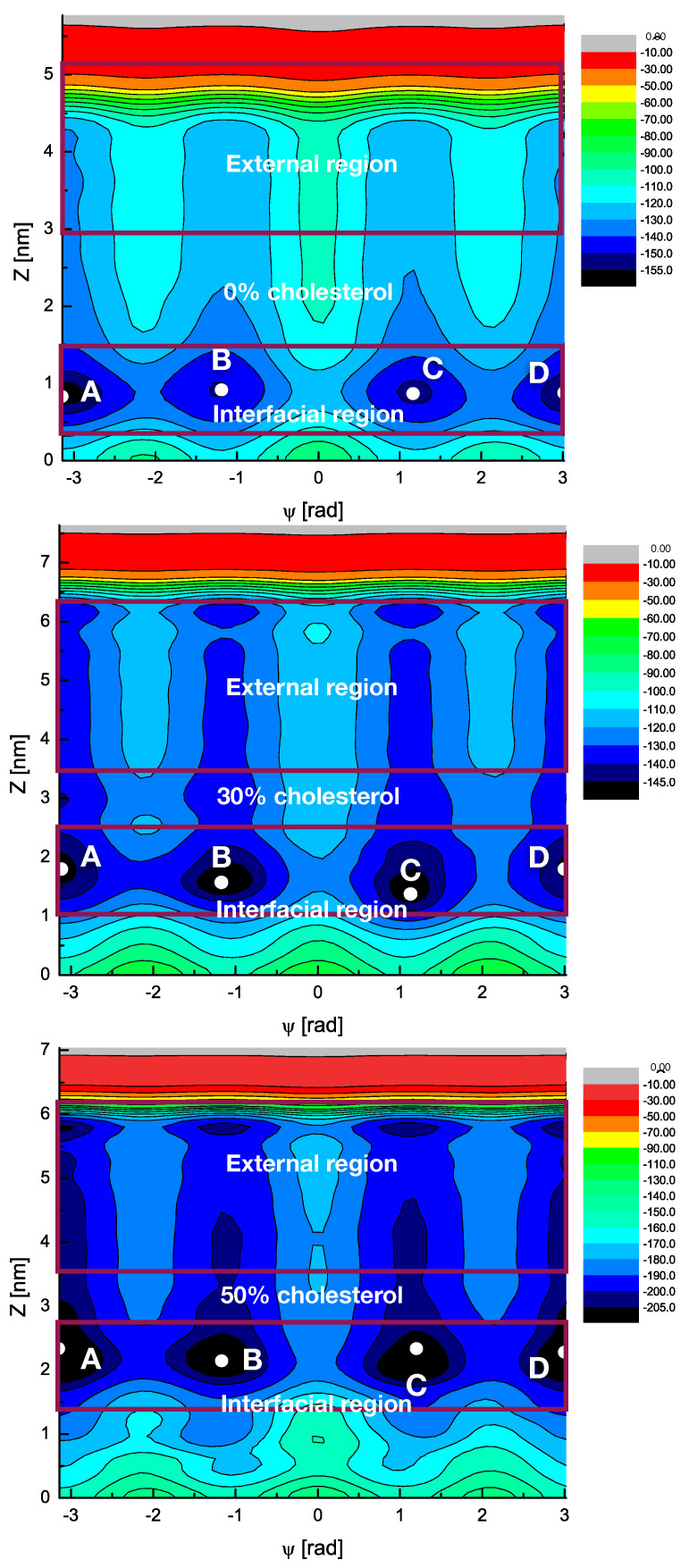
Two-dimensional (2D) free energy landscapes F(Ψ,z) (in kJ/mol) in the cholesterol-free case. Four stable state basins (A,B,C,D) are indicated.

**Table 1 ijms-22-02842-t001:** The averaged area per lipid (*A*) and thickness (Δz) of the anionic membrane for the systems studied in this work. Estimated errors in parenthesis. Data are partially taken from Refs. [107,108].

Small Molecule and Cholesterol Percentage	Phospholipid Species	*A* (nm${}^{2}$)	Δz (nm)
TRP-0%	DPPC	0.614 (0.008)	3.97 (0.05)
TRP-30%	DPPC	0.408 (0.002)	4.89 (0.04)
TRP-50%	DPPC	0.401 (0.002)	4. 78 (0.03)
SER-0%	DPPC	0.613 (0.015)	3.83 (0.05)
MEL-0%	DMPC	0.618 (0.005)	3.49 (0.06)
MEL-30%	DMPC	0.421 (0.007)	4.43 (0.03)
MEL-50%	DMPC	0.402 (0.008)	4.47 (0.03)

**Table 2 ijms-22-02842-t002:** Free energy barriers ΔF (in kJ/mol) for the binding of small molecules to DPPC.

Probe (Active Site)	O2-DPPC	O8-DPPC
H1 TRP	11.29	7.53
H2 TRP	8.02	4.18
H1 SER	7.95	6.53
H4 SER	7.45	7.87
H15 MEL	-	4.85
H16 MEL	8.03	1.97

**Table 3 ijms-22-02842-t003:** Free energy barriers ΔF (in kJ/mol) for the main transitions of a DMPC-bound MEL. Folded to extended corresponds to transitions between basins A and B or between C and D. Internal regions correspond to z∼0.

Cholesterol Percentage	Folded-Extended	Interface-Bulk	Interface to Internal Regions
0 %	18.8	25.3	40.2
30 %	19.7	14.1	50.7
50 %	17.6	9.1	55.5

**Table 4 ijms-22-02842-t004:** Self diffusion coefficients *D* (in 10${}^{-7}$ cm${}^{2}$/s) of TRP and MEL in systems with different cholesterol percentages. The estimated errors are in parenthesis.

Small Molecule	0% CHOL	30% CHOL	50% CHOL
TRP	3.48(0.80)	2.91(0.35)	14.0(0.2)
MEL	1.1(0.4)	3.9(0.6)	4.1(0.9)

**Table 5 ijms-22-02842-t005:** Characteristics of the molecular dynamics (MD) simulation runs performed in this work. The lengths of simulations include equilibration and production runs.

Phospholipids	Small Molecule	Waters	Total Length (ns)	Temperature (K)	Ion Pairs
204 DPPC	TRP	4962	800	323.15	17 Na${}^{+}$ + 17 Cl${}^{-}$
204 DPPC	SER	4962	800	323.15	17 Na${}^{+}$ + 17 Cl${}^{-}$
204 DMPC	MEL	10250	800	303.15	21 Na${}^{+}$ + 21 Cl${}^{-}$

**Table 6 ijms-22-02842-t006:** WTM simulation parameters.

Parameter	0%	30%	50%
Gaussian width of CV1 [nm]	0.30	0.30	0.25
Gaussian width of CV2 [degrees]	20	20	20
Starting (Gaussian) hill [kJ/mol]	1.0	1.0	1.0
Deposition stride [ps]	1	1	1
Bias factor	10	10	20
Simulation time [ns]	1100	1400	1400

## Data Availability

Data available in a publicly accessible repository.

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
