# Peer review of "Microscopic Interactions of Melatonin, Serotonin and Tryptophan with Zwitterionic Phospholipid Membranes"

_ijms, 2021, doi:10.3390/ijms22062842_

Round 1
Reviewer 1 Report
In the present manuscript the author perform an interesting computational study about the interaction of melatonin, serotonin and tryptophan with plasma membrane through simulation methods. The diffusion of these compounds has been studied and discussed in many studies. This manuscript contibute to understand the diffusion process of these compounds in phospholipid membranes. I would like to do some specific comments about the manuscript:
- The importance of melatonin in the skin is the protective effect against oxidative stress and damage, which is much more important than skin pigmentation. Lines 39-41 Rusanova et al. IJMS, 2019.
- Reference 41 is wrong, please correct it.
- Side effects of melatonin are controversal. Ref. 57 mentioned that melatonin treatment is safety even at high dose. After long time treatment, melatonin can show minor side effects but no one is important or serious. Lines 61-64 is ambiguous about the safety of melatonin, please correct it.
Author Response
We thank our referee for his/hers positive review. We have fully revised the manuscript and included changes to solve points raised by the referee.
Our answers are as follows:
- "The importance of melatonin in the skin is the protective effect against oxidative stress and damage, which is much more important than skin pigmentation. Lines 39-41 Rusanova et al. IJMS, 2019." Answer: We have rewritten lines 39-41 to focus on the action of melatonin related to aging processes such as oxidative damage and included the reference suggested. Changes marked in red in the "manuscript-revision-highlighted" file.
- "Reference 41 is wrong, please correct it." Answer: We have properly corrected spelling of names in Ref. 41 (now 43). Now they are correct.
- "Side effects of melatonin are controversal. Ref. 57 mentioned that melatonin treatment is safety even at high dose. After long time treatment, melatonin can show minor side effects but no one is important or serious. Lines 61-64 is ambiguous about the safety of melatonin, please correct it." Answer: We have rewritten lines 61-64 to stress the fact that melatonin is not dangerous for humans. Changes marked in red in the "manuscript-revision-highlighted" file.
Reviewer 2 Report
In this paper, the authors studied the interactions and binding mechanism between melatonin, serotonin and tryptophan with the cell membrane which comprised of DMPC or DPPC by molecular dynamics simulations. They also studied the effect of cholesterol on the interactions. The results quantitatively shows the behaviors of the three small molecules in the cell membrane and the difference in dynamics between them. Thus, I recommend publishing this paper in this journal.
Author Response
We thank our referee for his/hers positive evaluation. We have reviewed the full manuscript in order to improve the English spelling and style as well as the scientific explanation of results.
Reviewer 3 Report
In this article, the authors described the search review of the interactions and binding mechanisms between melatonin, serotonin, and tryptophan with the cell membrane structure at the atomic levels. The authors also presented their own research results obtained from simulation-based studies under several molecular components (such as cholesterol) of the lipid bilayer. As a result, the authors suggest the possible differences of the mechanisms whereby the three small molecules interact with the cell membrane, despite the molecules are similar with the chemical structure. This issue has progressively been focused worldwide in terms of pharmaceutical matter, and overall impact of their research is considered to be strong. My overall concern with the article describing the atomic relationship between lipid bilayer and small molecules is that information provided may offer something substantial that helps advance our understanding of possible delivery effect(s) of small molecules when they pass through the cell membrane as medicinal products. I recommend that this manuscript is acceptable for publication in the IJMS as it is.
Author Response

(The authors gave the same response as above.)
